# Investigating Biochar-Derived Dissolved Organic Carbon (DOC) Components Extracted Using a Sequential Extraction Protocol

**DOI:** 10.3390/ma15113865

**Published:** 2022-05-28

**Authors:** Hui Liu, Baowei Zhao, Xin Zhang, Liujun Li, Yue Zhao, Yingquan Li, Kaixiang Duan

**Affiliations:** 1School of Environmental and Municipal Engineering, Lanzhou Jiaotong University, Lanzhou 730000, China; liuhui@mail.lzjtu.cn (H.L.); 1320014@stu.lzjtu.edu.cn (X.Z.); 0217095@stu.lzjtu.edu.cn (L.L.); 0217114@stu.lzjtu.edu.cn (Y.Z.); 2Gansu Dust Suppression for Transportation and Storage Engineering Research Center, Lanzhou 730000, China; yingquanli@mail.lzjtu.cn (Y.L.); 022412125@hncj.edu.cn (K.D.)

**Keywords:** biochar, DOC concentration, UV–vis analyses, PARAFAC analysis

## Abstract

Biochar-derived dissolved organic carbon (DOC), as the most important component of biochar, can be released on farmland, improving fertility and playing a role in soil amendment and remediation. The complexity of molecular structures and diversity of DOC compounds have influenced these functions to some extent. A sequential extract protocol consisting of water (25 °C), hot water (80 °C), and NaOH solution (0.05 M) was used to fully extract DOC compounds and gain a thorough understanding of the possible DOC components released from biochar. Rape straw (RS), apple tree branches (ATB), and pine sawdust (PS) were pyrolyzed at 300, 500, and 700 °C, respectively, to make nine distinct biochars. A TOC analyser, ultraviolet-visible spectroscopy (UV–vis), and excitation–emission fluorescence (EEM) spectrophotometer were used in conjunction with parallel factor analysis (PARAFAC) to determine the distribution of DOC content, the diversity of aromaticity, molecular weight characteristics and components of biochar-derived DOC. The results show that the relative distribution of water-extractable fractions ranged from 3.21 to 35.57%, with a low-aromaticity and extremely hydrophilic fulvic-acid-like compounds being found in the highest amounts (C2 and C3). The smallest amount of hot water-extractable components was produced from the release of small-molecule aliphatic compounds adsorbed on biochar and susceptible to migration loss once in a soil solution. More than half of the biochar-derived DOC was released in a NaOH solution, which primarily consisted of humic-acid-like compounds (C1), with higher molecular weights, more aromaticity, and lower bioavailability, according to the distribution of DOC in various extractants. In addition, the pyrolysis temperature and biomass type had a significant impact on the DOC properties released by biochar. As a result, the findings of this study showed that using a sequential extract protocol of water, hot water, and NaOH solution in combination with spectroscopic methods could successfully reveal the diversity of biochar-derived components, which could lead to new insights for the accurate assessment of potential environmental impacts and new directions for biochar applications.

## 1. Introduction

Carbon sequestration [1,2], farmland fertility enhancement [3], and soil remediation have all received a lot of attention from the environmental and agricultural industries. Biochar is a black carbon-rich solid generated by the thermal decomposition of biomass under oxygen-limited circumstances at temperatures ranging from 300 to 700 °C [4]. Once applied to the soil, dissolved organic carbon (DOC), the most mobile and active component of biochar, can be rapidly released into soil water via soil pore water, irrigation, and rain [5]. It can immediately modify the physicochemical characteristics of soil, increase the organic matter content of soil [6,7], impact the microbial community structure [8,9], and change the soil organic pollutants and heavy metal migration. More crucially, these functionalities are supposedly determined by the complexity of molecular structures and the variety of biochar-derived DOC compounds. For example, combining cacao-shell-derived biochar with acrisols at 10 wt.% resulted in a 15-fold increase in soil DOC content and changed the DOC composition towards molecules with a larger size and higher aromaticity [10]. The aliphatic and high-aromatic combustion-derived condensed polycyclic aromatic chemicals in biochar-derived DOC have a specific influence on the quantity and population of the particular microorganisms [11]. Two-dimensional correlation spectroscopy demonstrated that the phenolic, carboxyl groups, protein-, and fulvic-like substances of biochar-derived DOC had varied binding capacities with heavy metals, which directly altered the transportation and transformation of heavy metals in soil [9]. As a result, a full understanding of the potential DOC components released from biochar is needed for an appropriate assessment of the influence of biochar on the applied soil.

The composition, chemical structure, and functional characteristics biochar-derived DOC are determined by the raw materials and pyrolysis conditions [12,13,14,15,16]. In general, a lower DOC content of woody biochars (oak wood, pine wood, Chinese bamboo, etc.) compared to herbaceous biochars (corn stover, soybean, switchgrass, sugarcane bagasse, etc.) and manure biochars (bull manure, dairy manure, poultry manure, etc.) could be a result of their compositional differences [12]. Wood-based feedstocks include a greater proportion of lignin, which is more thermally stable and rapidly generates fixed carbon rather than DOC [17]. In contrast, other feedstocks contain a greater proportion of hemicellulose and cellulose, which are more lead to DOC formation. Smith et al. discovered that the unique carbohydrate ligneous components and sulfur-containing condensed ligneous components are present solely in the extract solution of pine woody biochar, but not in the extract solution of any other biochars (peanut shell or chicken litter biochar) [14]. Additionally, Gui et al. found that when the pyrolysis temperature increased from 200 to 700 °C, the DOC content of manure biochar (chicken, swine, and dairy) abruptly reduced to a steady level, but the humic-acid-like compounds increased until 400 °C and then gradually decreased [18]. Lin et al. discovered that the DOC concentration of biochar decreased as the pyrolysis temperature increased from 450 to 550 °C. The fraction of DOC changed from a humic-like substance and low-molecular-weight neutrals (including low-molecular-weight neutrals, alcohols, aldehydes, ketones, and surges, etc.) to low-molecular-weight acids (protic organic acids) at low pyrolysis temperature [19].

Additionally, the extraction solution’s physicochemical parameters influenced the amount and composition of DOC released from biochar. DOC produced from charcoal may be extracted conventionally using an acid (0.1 M HCl), base (0.1 M NaOH), deionized water, and hot water (60 or 80 °C). Liu et al. extracted DOC components from 46 biochars using deionized water, acid (0.1 M HCl), and base (0.1 M NaOH) solutions. They discovered that the concentration, aromaticity, and molecular weight of biochar-derived DOC generally increased in the order of acid-extractable DOC, water-extractable DOC, and base-extractable DOC samples [12]. Li et al. used EEM spectrophotometry with PARAFAC and UV–vis spectroscopy to determine the amount of dissolved organic matter (DOC) released by wheat-straw-derived biochar in various extracts (deionized water; 0.1 mol/L HCl; 0.1 mol/L NaOH; 0.1 mol/L NaCl) at various temperatures (20 and 60 °C). The results indicate that the highest levels of DOC release occurred under high temperatures. The humification index (HIX) of DOC released from biochar in different extracts was likewise greater at 60 °C (1.31–1.92) than at 20 °C (0.99–1.09) [20]. Wu et al. investigated the quantity and quality of dissolved organic carbon (DOC) released from wetland-plant-derived biochar under various extracting conditions (deionized water, 0.1 mol/L HCl, 0.1 mol/L NaOH, and 0.1 mol/L NaCl), and the result showed that the highest DOC content was released in the alkaline solution and lowest in the deionized water [21]. The difference of DOC characteristics in different extraction solutions indicates the diversity of DOC components, which is strongly dependent on the physicochemical properties of the extraction solution. A specific extraction solution can only extract certain DOC compounds from biochar. Thus, drawing quantitative and qualitative conclusions about the diversity of compounds and molecular characteristics of all potentially releasable biochar-derived DOC fractions based on water-, hot water-, salt-, acid- or alkali-extractable components alone is problematic. To address this issue, Tfaily et al. utilized water, followed by methanol–chloroform or water, followed by acetonitrile and a CHCl_3_ sequential extraction procedure to extract organic compounds from soil and sediments. The results show that the sequential protocol has a higher sensitivity and allows the diversity of organic matter (OM) to be revealed in different terrestrial ecosystems [22]. Zhu et al. established a 4-step sequential extraction protocol to analyse carbohydrates in aquatic sediments, and the results show that recoveries of neutral sugars, amino sugars and sugar alcohols are on average ~60% higher than with a reference one-step extraction method [23]. Ellerbrock et al. used a water and 0.1 M sodium–pyrophosphate solution extraction protocol to isolate and characterize organic matter (OM) fractions with varying degrees of solubility [24]. They discovered that the sodium–pyrophosphate soluble fraction was more stable than the water-soluble fraction and demonstrated that sequential extraction appeared to be useful for isolating (OM) fractions with varying degrees of stability. However, few studies have fully documented the extraction of the DOC generated from biochar using a stepwise process. It is critical to create a sequential extraction strategy based on the likely release sequence of DOC components to achieve a thorough extraction of DOC.

Thus, to thoroughly extract the DOC fraction from biochar, this study designed a stepwise extraction process consisting of water (at room temperature), hot water (80 °C), and an alkali solution (0.05 M NaOH). The severe conditions (80 °C and 0.05 M NaOH) were chosen to achieve complete DOC release from biochar and to better forecast the possible released DOC components if biochar was exposed to natural settings for an extended period. Therefore, nine biochar samples were obtained at 300, 500, and 700 °C pyrolysis using rape straw (RS), apple tree branches (ATB), and pine sawdust (PS), respectively. Rape straw, apple tree branches and pine sawdust were selected as the feedstocks because they are widely distributed agricultural and forestry waste in China [25,26,27]. The DOC of matching biochars was successively extracted using water (at room temperature), hot water (80 °C), and a NaOH solution (0.05 M). Quantitative and qualitative analyses of the obtained extracts were performed using a TOC analyser, UV–vis spectroscopy, and EEM combined with PARAFAC. The objective is to completely extract DOC from biochar and to analyse its components and chemical structure in detail. When biochar is employed as an amendment, these findings will be critical for predicting potential effects on agricultural and soil environments. 

## 2. Materials and Methods

### 2.1. Biochars

Rape straw (*Brassica campestris* L.) was harvested from farmland in Wudu, Gansu Province, China. Apple tree branches (*Malus pumila Mill.*) with a length ranging from 10 to 20 cm, were collected from the pruned branches of the farmer’s orchard in Lixian, Gansu Province, China. Pine sawdust (*Pinus bungeana Zucc. et Endi*) was obtained from Lanzhou Timber Market, Gansu Province, China. The basic composition of samples is shown in Table 1. All samples were dried in a furnace (202-2, Beijing Linmao Technology Company, Beijing, China) at 105 °C for 24 h, then pulverized into a fine powder using a pulveriser (9FZ-15, Shanxi Mingqingda Instrument, Yuncheng, China), passed through a 40-mesh sieve, and stored in a brown jar with lid. The appropriate biomass was placed into muffle furnace (SX2 series, Shanghai Jianzhan Instrument, Shanghai, China) for pyrolysis under N_2_ atmosphere with a heating rate of 10 °C·min^−1^ to a final temperature of 300, 500, and 700 °C, respectively, and held for 6 h at peak temperature. After complete cooling, biochar samples were stored in an airtight glass container and named by feedstock and pyrolysis temperature: RS 300 for rape straw pyrolyzed at 300 °C, ATB 300 for apple tree branches pyrolyzed at 300 °C, and PS 300 for pine sawdust pyrolyzed at 300 °C.

Proximate analyses of feedstocks were conducted by determining the content of moisture, ash, volatile matter, and calculating the content of fixed carbon by weight difference. Moisture content was determined by placing the samples into glass dishes that were dried at 105 °C to a constant weight. The volatile matter was determined by keeping the samples in a porcelain crucible in a muffle furnace at 900 ± 10 °C for 7 min. Ash content was obtained by keeping samples in a porcelain crucible in a muffle furnace at 700 °C for half of an hour. The lignocellulosic content of samples, which included hemicellulose, cellulose, and lignin was obtained based on the methods provided by Li et al. [28]. All the experimental determinations were repeated three times.

### 2.2. DOC Sequential Extraction Experiments

Three different types of fluids were employed to extract DOC from biochar consecutively, namely water, hot water (80 °C), and 0.05 M NaOH. Three steps were included in the detailed extraction procedure: (1) In a 500 mL Erlenmeyer flask, 3 g of biochar was combined with 300 mL of deionized water and then oscillated at 25 °C and 180 rpm for 16 h on a bath reciprocal shaker (CHA-S, Jiangsu Jingyu instrument, Suqian, China). The supernatant was filtered via a 0.45 μm membrane and collected in brown bottles labelled as DOC water-extractable. For the next procedures, residual biochar was collected and dried for 24 h at 105 °C.

The dried residual biochar was weighed and mixed with 80 °C deionized water (1:100 *w*/*v*) in a 500 mL Erlenmeyer flask. The flask was then oscillated in an 80 °C bath for 16 h. The supernatant was filtered via a 0.45 μm membrane and collected in brown bottles labelled as DOC extractable with hot water. For the subsequent extraction, the leftover biochar was dried in an oven at 105 °C for 24 h.

The dry residual biochar was weighed and mixed in a 500 mL Erlenmeyer flask with 0.05 M NaOH at a ratio of 1:100 (*w*/*v*), then oscillated at 25 °C and 180 rpm for 16 h. The resulting supernatant was labelled as extractable DOC from a NaOH solution. All samples underwent three parallel trials and were kept at 4 °C during the test in a refrigerator. 

### 2.3. Concentration and UV–vis Analyses of Biochar-Derived DOC

The TOC concentrations extracted in water, hot water, and NaOH solution were determined with TOC/TN analyser (MULTI N/C 2100, Analytik Jena AG, Jena, Germany), representing the total organic carbon (TOC) concentration in different extraction fluids. DOC concentration of the biochar (mg·g^−1^) was calculated using the following Equation (1):(1)DOC concentration=V×C/M
where *V* is the volume (*L*) of extraction solution, *C* is the TOC concentration (mg·L^−1^) in the extraction solution, and *M* is the mass of bulk biochar (g). The cumulative DOC concentration was the sum of the water, hot water, NaOH solution-extractable DOC concentration.

The absorbance of the extraction solution was measured using a UV–visible spectrophotometer (UV-3600, Shimadzu, Kyoto, Japan) within a spectrum of 200 to 750 nm, at a 1 nm increment. The spectral absorption ratio of 254 and 365 nm (*E*_2_/*E*_3_) was calculated using the following Equation (2):(2)$$E_{2}/E_{3}=A_{254}/A_{365}$$
where $A_{254}$ and $A_{365}$ are the absorbance at 254 and 365 nm.

The specific UV absorption at 254 nm (*SUVA*_254_) was calculated using the following Equations (3) and (4):(3)$$a_{254}=2.303\;A_{254}/L$$
(4)$$SUVA_{254}=a_{254}/\left(L\times C\right)$$
where $a_{254}$ is the absorbance coefficient measured at 254 nm (m^−1^), and *L* is the length of quartz cuvette, which is 1 cm in this experiment. *SUVA*_254_ is the specific UV absorption at 254 nm (L·mg^−1^·m^−1^)

### 2.4. EEM Fluorescence and PARAFAC of DOC

The EEM fluorescence of each extract was determined using a fluorescence spectrophotometer (Hitachi, F-7100, Tokyo, Japan), using a 1 cm quartz cuvette at 5 nm increments, a 0.5 s response time, and 2400 nm min^−1^ scan speed. Milli-Q ultra-pure water was used as a blank EEM to subtract Raman scattering effects from biochar extract samples. The obtained EEM data were analysed using PARAFAC modelling in MATLAB R2010a, as described by Stedmon and Bro [29]. Based on PARAFC process for analysing the EEM fluorescence of the samples, once the PARAFAC model identified the components, the positions of the fluorescence peaks at different excitation and emission wavelengths, and the corresponding fluorescence intensities of the components contained in the sample, were automatically output. Relative proportions of different components could be calculated based on their fluorescence intensity.

### 2.5. Statistical Analysis

Each experiment was replicated three times, and the findings were provided as means ± standard deviations. SPSS 23.0 was used to conduct the statistical analysis. One-way and two-way analyses of variance were conducted to determine if there were any significant differences in the acquired data, and all significant levels were 0.05 (*p* < 0.05).

## 3. Results and Discussion

### 3.1. Distribution of DOC Content in Water, Hot Water and NaOH Solution

Table 2 summarizes the DOC concentrations of biochars progressively extracted using water, hot water, and NaOH solution. When DOC concentrations in different extraction solutions were compared, it was discovered that the NaOH solution-extractable DOC concentration was considerably (*p* < 0.05) greater than that of water and hot water, regardless of the feedstocks and pyrolysis temperatures. According to the distribution of DOC components in various extraction solutions, as seen in Figure 1, the relative fraction of DOC extracted in NaOH ranged between 55.46 and 93.21%. Previous studies showed that the majority of the DOC components in biochar may be released only in alkaline circumstances [12,20]. It is possible that chemical treatment with NaOH or KOH promoted the dissociation of carboxyl, phenyl groups and the breaking of ester bonds, resulting in a rise in DOC concentration [30]. Thus, when biochar is employed as a soil supplement to increase the organic matter content of the soil, the pH of the soil (especially for soda saline-alkali soil) may play a significant role in the amendment’s efficacy [31,32,33].

Between 3.21 and 35.57% of water extractable fractions were found to have a low relative distribution. These components are the most labile part of biochar-derived DOC and may have various consequences on the environment once released. The relative abundance of hot water extracted DOC components was the lowest, ranging from 3.57 to 14.81%. Most of the organic content adsorbed on minerals in biochar was released as hot-water-extractable fractions [22]. These fractions are mostly carbohydrates, phenols, lignin monomers, and nitrogen-containing chemicals [34,35] and exhibit a stronger biodegradability than cold water extracts [36].

Additionally, the amount of DOC released in each extraction solution is regulated by the kind of feedstocks used and the pyrolysis temperatures. Under the same pyrolysis temperature, the cumulative DOC concentrations of RS 300, RS 500, and RS 700 were 49.17 ± 0.95, 5.68 ± 0.33, and 4.7 ± 0.24 mg·g^−1^, about 2.7, 2.5, and 6.2 times those of ATB 300, ATB 500, and ATB 700 and 6.2, 7.2 and 6.4 times those of PS 300, PS 500, and PS 700, respectively (Table 2). The DOC concentrations of RS biochar extracted with water, hot water, and NaOH solution are similarly much greater than those of ATB and PS. 

RS is a herbaceous biomass with a higher proportion of hemicellulose and cellulose (as shown in Table 1), which promotes the synthesis of volatile organic compounds and bio-oil inside the biochar pore structure by recondensation and trapping, which may then be released as DOC [37]. ATB and PS are woody materials that include a higher concentration of lignin components that are more difficult to breakdown, resulting in pyrolysis products with a higher solid content and a lower DOC content [38]. Herbaceous plants (e.g., RS) have a greater ash content, catalysing the thermochemical reaction of lignocelluloses, resulting in increased DOC content [39]. Notably, when the pyrolysis temperature was increased from 300 to 700 °C, the cumulative DOC concentrations of RS, ATB, and PS biochar decreased from 49.17 ± 0.95, 18.11 ± 0.55 and 7.88 ± 0.26 mg·g^−1^ to 4.7 ± 0.24, 0.76 ± 0.37 and 0.74 ± 0.09 mg·g^−1^, respectively. Increased pyrolysis temperature accelerated the carbonization process, resulting in the continuous decomposition, condensation, cyclisation, and polymerization of the labile organic compounds formed at lower pyrolysis temperatures and a decrease in DOC concentrations.

Interestingly, the impact of pyrolysis temperature on the extractable fractions of biochar was not uniform. The relative quantities of water- and hot-water-extractable fractions of RS biochar declined as the pyrolysis temperature rose, but the NaOH solution extractable fractions increased (Figure 1). The quantities of water, hot water, and sodium hydroxide in ATB and PS biochar were completely opposed to those in RS biochar. This may also imply differences in the chemical structure of DOC components extracted from RS, ATB, and PS biochar in water, hot water, and NaOH solution as the pyrolysis temperature increases.

### 3.2. Aromatic Abundance and Molecule Characteristics of DOC in Water, Hot Water and NaOH Solution

The aromatic abundance and molecular properties of biochar-derived DOC components released into water, hot water, and sodium hydroxide solutions were determined using the *SUVA*_254_ value [20] and *E*_2_/*E*_3_ [12] ratio of the UV–vis absorption spectra. According to Table 3, the *SUVA*_254_ value of RS biochar-derived DOC in water, hot water and NaOH decreased from 4.51 ± 0.10 to 0.21 ± 0.02 L·mg^−1^·m^−1^, 4.53 ± 0.11 to 1.67 ± 0.14 L·mg^−1^·m^−1^, and 6.77 ± 0.20 to 0.30 ± 0.02 L·mg^−1^·m^−1^, respectively, increasing the pyrolysis temperature from 300 to 700 °C. Generally, the *SUVA*_254_ value was positively correlated with the aromatic of DOC [40]. A value > 4 L·mg^−1^·m^−1^ represents more aromatic hydrophobic substances in DOC, and a value < 3 L·mg^−1^·m^−1^ represents more hydrophilic substance in DOC [41]. It is shown that a progressive decrease in the aromaticity and hydrophobicity of RS-biochar-derived DOC occurred as the pyrolysis temperature increased, whereas the hydrophilicity increased, regardless of the extraction solution. The DOC portion of biochar generated at a low temperature (300 °C) increased hydrophobicity due to the presence of aliphatic compounds and disappeared beyond 500 °C due to aliphatic component volatilisation [42]. The hydrophobic component of DOC was highly preferred over soil binding, resulting in the decreased DOC-induced transport of metals and organic contaminants but increased sorption to the mineral phase [43]. Additionally, once the hydrophobic organic matter component of biochar entered the soil, it promoted the long-term stability of soil aggregates due to the formation of organic matter–soil complexes with hydrophobic characteristics [44]. The DOC portion of RS biochar generated at a high temperature (700 °C) demonstrated the increased the co-sorption and co-transport of hydrophilic pollutants, which affect the amount of metals, and organic pollutants in the soil solution [45]. *SUVA*_254_ values for ATB and PS-biochar-derived DOC in water, hot water, and sodium hydroxide solution increased from 300 to 500 °C and subsequently declined. This is consistent with prior research, which established a key inflection point for the thermal formation/degradation of carboxyl functional groups around 500 °C [46]. The *SUVA*_254_ value for the water-, hot-water- and NaOH-solution-extracted DOC components of ATB and PS biochar were all less than 4.0 L·mg^−1^·m^−1^, except PS 500. Additionally, these results implied that the DOC components released from ATB and PS biochars were less aromatic and more hydrophilic, making them more prone to loss via off-site transport. The mean value of *SUVA*_254_ of DOC released in the NaOH solution (3.21 ± 0.28 L·mg^−1^·m^−1^) was significantly higher (*p* < 0.05) than in water (2.46 ± 0.24 L·mg^−1^·m^−1^) and hot water (2.56 ± 0.22 L·mg^−1^·m^−1^). At the same time, there was no significant difference between water and hot water. It is assumed that the presence of OH^−^ in the solution increased the dissociation of carboxyl and phenyl groups or the breaking of ester bonds and caused the dissolution of macromolecules [30], resulting in a higher aromaticity of DOC components and low mobility. The decreased aromaticity and increased hydrophilicity of the water- and hot-water-derived DOC components make them more mobile and bioavailable.

The *E*_2_/*E*_3_ ratio of RS biochar-derived DOC in water, hot water, and NaOH ranged from 5.34 ± 0.07 to 9.50 ± 0.21, 3.91 ± 0.09 to 34.00 ± 0.33, and 2.81 ± 0.02 to 9.56 ± 0.78, respectively, increasing the pyrolysis temperature from 300 to 700 °C. Because the soluble organic carbon of macromolecules showed increased aromaticity at longer wavelengths, the *E*_2_/*E*_3_ ratio was reversely associated with molecular weight and size in samples [47]. This revealed that greater pyrolysis temperatures resulted in lower-molecular-weight DOC components in RS biochar. Because pyrolytic reactions occur across a wide temperature range, the initially generated organic matter undergoes various secondary processes, such as decomposition, condensation, cyclisation, and polymerisation, resulting in the formation of low-molecular-weight compounds [39]. Lin et al. discovered an increase in low-molecular-weight acids (protic organic acids) and neutrals (low-molecular-weight alcohols, aldehydes, ketones, sugars, and so on) in biochars formed at higher temperatures as a result of pyrolysis secondary reaction [19]. Except for PS 700, the *E*_2_/*E*_3_ ratio of hot water extractable DOC components from ATB and PS biochar increased with the pyrolysis temperature but those of water and NaOH solution exhibited the reverse trend. This means that larger molecular components of high-temperature biochar were liberated into water and the NaOH solution. According to previous research, a greater pyrolysis temperatures accelerate lignin degradation in ATB and PS, resulting in more complex phenolic compounds with higher molecular weights [48] and polynuclear aromatics [49]. By and large, the greater the molecular weight and size of DOC components, the greater their bioavailability [50]. This indicates that the bioavailability of ATB and PS biochar DOC extracted with water and NaOH solution increased as the pyrolysis temperature increased. Nonetheless, the hot water extraction fraction had the reverse tendency. The mean *E*_2_/*E*_3_ ratio of DOC released in hot water (10.31 ± 0.46) was significantly higher (*p* < 0.05) than in water (6.06 ± 0.44) and NaOH (4.23 ± 0.35). This revealed that the molecular size of biochar-derived DOC released in hot water was much smaller than that released in water and NaOH solution, which might be due to hot water favouring the release of smaller aliphatic DOM molecules that were more tightly bound to biochar [10].

### 3.3. Fluorescence Characteristic and Compositions of DOC

In this study, four fluorescence components of biochar-derived DOC samples were found, and their associated excitation/emission loadings are presented in Figure 2. Component 1 (C1), in particular, displayed a single peak with a maximum excitation at 240 nm and a maximum emission at 400 nm. It was identified as a humic-acid-like substance [51], with increased aromaticity and decreased solubility [52,53]. Components 2 (C2) and 3 (C3) were characterized as having a fulvic-like structure [54]. C2 exhibited two peaks with maximal excitations at 208 and 300 nm and a maximum emission at 418 nm. Additionally, the C3 component exhibited two peaks with a maximum excitation at 230 and maximum emission at 460 nm. C3 had a longer emission wavelength than C2, indicating that it contained more aromatic, conjugated, and high-molecular-weight compounds [55]. C4 exhibited two peaks with excitation maxima at 218 and 260 nm and a maximum emission of 325 nm, indicating that it was classified as a protein-like molecule [56] and was frequently associated with biological activity in natural waters [57].

To better understand the fluorescence differences between biochar-derived DOC components extracted in water, hot water, and NaOH solution, the relative distributions of four PARAFAC components were calculated in detail. The distribution of four PARAFAC components in DOC was found to vary significantly amongst extractant solutions. As seen in Figure 3, the C1 component of DOC generated from RS, ATB, and PS biochar varied between 26.43 and 37.53%, 23.64 to 28.24%, and 8.01 to 19.60%, respectively. Compared to woody-based (ATB and PS) biochar, herbaceous-based (RS) biochar released more humic-like compounds into the water. This may be related to the conversion of the abundant random and amorphous fractions of hemicellulose and cellulose to aromatic fractions. Except for ATB 500, C1 had a maximum value at 500 °C, implying that C1 component initially increased and subsequently decreased as the pyrolysis temperature climbed. The humic-like substance was generated by the oxidation of free primary hydroxyls at temperatures less than 500 °C and was then consumed by dehydrogenation and dehydroxylation processes at temperatures greater than 500 °C [46]. The distributions of C1 in NaOH solution were 26.75–44.94% for RS, 37.24–67.58% for ATB, and 32.81–53.35% for PS, respectively, which were much greater than the distribution in water and hot water extractant. The reason for this might be that the humic-like material has a higher molecular weight and a higher hydrophobicity, which results in a poorer solubility in both cold and hot water [58]. Additionally, the presence of OH-(NaOH) may decrease the hydrogen connection between the humic-like substance and the biochar, encouraging a humic-like substance release from the biochar. Thus, when biochar is applied as a soil supplement to enhance soil organic matter (SOM), more humic-like substances are released into the soil due to the shift in soil pH, which benefits soil fertility and mineral nutrient bioavailability and uptake both directly and indirectly [59].

The distribution of fulvic-like substances (C2 + C3) from RS, ATB and PS biochar ranged between 54.72 and 61.41%, 42.68 and 65.24%, and 65.97 and 75.97% in water; 58.25 and 77.3%, 42.41 and 74.19%, and 78.87 and 83.54% in hot water; and 54.61 and 73.24%, 29.68 and 46.24%, and 42.16 and 52.86% in the NaOH solution, respectively. The DOC components of water and hot water were clearly seen to be dominated by fulvic-like compounds. This is consistent with the findings of Li et al., who found that the dissolved organic matter composition released by wheat-straw-derived biochar in deionized water and an acidic extractant was dominated by a compound similar to fulvic acid [20]. Fulvic-like substances have a low molecular weight, including some (poly)phenolic and other aromatic structures, and are frequently soluble in water [60]. Additionally, when considering the total pyrolysis temperature, the fulvic-like compounds released in water and hot water from biochar generated from various raw materials were in the order ATB~PS > RS. According to previous research, between 25% and 30% of the aromatic carbon in fulvic acid is due to lignin breakdown [61]. As a result, biochar made from lignin-rich materials (ATB and PS) included a higher concentration of fulvic-like compounds released into water and hot water. Furthermore, unlike humic-like chemicals, fulvic-like substances were a more active group constituent in DOC [62]. Due to the strong metal blinding sites in fulvic acidic, stable chelates of fulvic acidic and metal may form [63], which may play a more substantial role in heavy metal remediation.

Except for ATB 500, the fraction of C4 released from RS, ATB, and PS biochar is less than 20% in water, hot water, and NaOH solution. This demonstrates that trace amounts of nitrogen-containing compounds, as well as a part of heterocyclic nitrogen compounds that are insoluble in water, are present in bulk biochar [64,65]. These trace levels of protein-like components might supplement the bioavailability of dissolved organic nitrogen (DON), hence increasing plant production and quality [66]. By comparing the relative proportions of C4 components released from different biochars, it was discovered that the C4 content of ATB and PS biochars was between 2.75 and 33.68%, 4.22 and 17.17%, respectively, significantly higher than the C4 content of RS biochar, which was between 0 and 14.42%. Additionally, regardless of the extract solution, the protein-like component released from ATB and PS initially decreased and subsequently increased with the increasing pyrolysis temperature. This is consistent with Uchimiya’s [67] conclusion that the formation of protein-like substances is dependent on lignin decomposition, which occurs primarily at temperatures between 300 and 480 °C (chain fragmentation to form monomeric phenol units) and above 500 °C (thermal cracking of tar residue in the solid sample) [68]. This also means that lignin-rich materials such as biochar produced a greater amount of nitrogen-containing compounds, which had the potential to offer economic and environmental value when utilised as a slow-release nitrogen fertiliser [69].

## 4. Conclusions

The results indicate that there were substantial changes in the amount and composition of DOC extracted successively from water, hot water, and NaOH solution. Water extractable DOC components are mostly composed of low aromaticity and extremely hydrophilic fulvic acid-like substances (C2 and C3), which are the most labile and diversified components with potentially complicated impacts on the application environment. The components of hot water extracts are composed of small molecule aliphatic compounds that are prone to migration loss once in soil solution. The largest extractable DOC concentration is found in the NaOH solution, predominantly constituted of humic-acid-like substances (C1) with considerably larger molecular weights and stronger aromaticity. Additionally, the DOC component formed from biochar was impacted by the raw material used to prepare the biochar and the pyrolysis temperature. Thus, to appropriately analyse the potential environmental impact of biochar on the cropland or soil on which it is used, it is necessary to completely extract and describe the DOC components produced by biochar and choose the optimal biomass and pyrolysis temperature.

## Figures and Tables

**Figure 1 materials-15-03865-f001:**
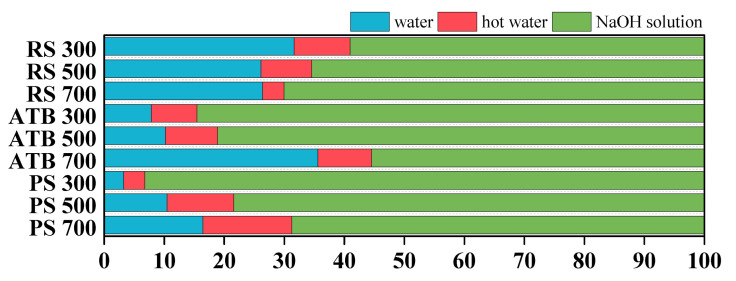
The distribution of DOC concentration in water, hot water and NaOH solution.

**Figure 2 materials-15-03865-f002:**
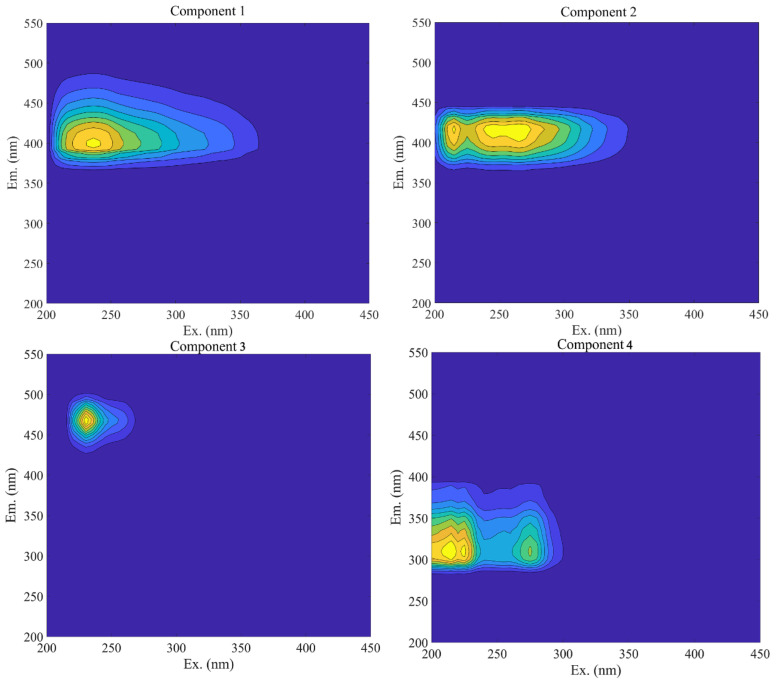
EEM contours for the four major fluorescent components identified by PARAFAC analysis.

**Figure 3 materials-15-03865-f003:**
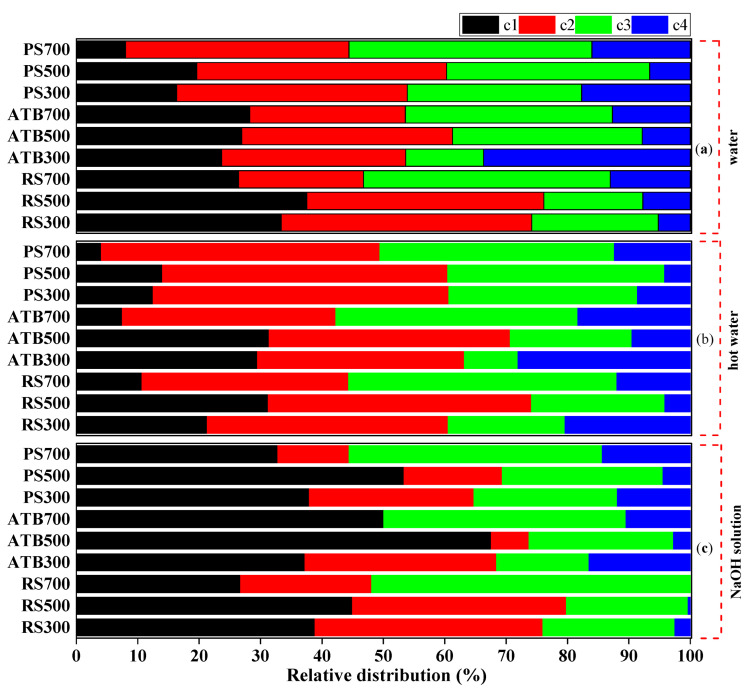
The relative distribution of EEM-PARAFAC analysis components in water (**a**), hot water (**b**) and NaOH solution (**c**).

**Table 1 materials-15-03865-t001:** The basic composition of samples (*n* = 3).

Composition		Rape Straw	Apple Tree Branches	Pine Sawdust
Proximate Analysis	water (%)	5.01 ± 0.08	2.63 ± 0.12	2.24 ± 0.33
	ash (%)	3.49 ± 0.03	2.32 ± 0.02	1.55 ± 0.02
	volatile (%)	80.02 ± 0.54	80.00 ± 0.42	82.72 ± 0.39
	fixed carbon (%)	11.47 ± 0.23	15.04 ± 0.17	13.49 ± 0.16
Lignocellulosic contents	cellulose (%)	41.08 ± 2.26	46.22 ± 1.30	54.69 ± 2.01
	hemicellulose (%)	27.95 ± 5.13	16.68 ± 0.55	11.08 ± 2.65
	lignin (%)	16.00 ± 3.08	30.73 ± 0.97	26.43 ± 1.22

**Table 2 materials-15-03865-t002:** DOC concentrations (mg·g^−1^) of RS, ATB and PS biochars released in the different extraction solutions (*n* = 3).

Biochar	DOC Concentration (mg·g^−1^)			Cumulative DOC Concentration (mg·g^−1^)
	Water	Hot Water	NaOH Solution	
RS 300	15.55 ± 0.09 b	4.59 ± 0.22 c	29.03 ± 0.64 a	49.17 ± 0.95
RS 500	1.48 ± 0.07 b	0.48 ± 0.05 c	3.72 ± 0.21 a	5.68 ± 0.33
RS 700	1.24 ± 0.12 c	0.17 ± 0.03 b	3.29 ± 0.09 a	4.70 ± 0.24
ATB 300	1.42 ± 0.20 b	1.38 ± 0.07 b	15.31 ± 0.28 a	18.11 ± 0.55
ATB 500	0.23 ± 0.03 b	0.20 ± 0.06 b	1.85 ± 0.06 a	2.28 ± 0.15
ATB 700	0.27 ± 0.03 b	0.07 ± 0.02 c	0.42 ± 0.08 a	0.76 ± 0.11
PS 300	0.25 ± 0.02 b	0.28 ± 0.08 b	7.35 ± 0.16 a	7.88 ± 0.26
PS 500	0.08 ± 0.01 b	0.09 ± 0.02 b	0.62 ± 0.02 a	0.79 ± 0.05
PS 700	0.12 ± 0.03 b	0.11 ± 0.03 b	0.51 ± 0.03 a	0.74 ± 0.09

Different letters in the same line indicate significant differences (*p* < 0.05) among water, hot water and NaOH solution.

**Table 3 materials-15-03865-t003:** The *SUVA*_254_ (L·mg^−1^·m^−1^) and *E*_2_/*E*_3_ ratio of DOC released in water, hot water and NaOH solution (*n* = 3).

Biochar	*SUVA*_254_ (L·mg^−1^·m^−1^)			*E*_2_/*E*_3_		
	Water	Hot Water	NaOH Solution	Water	Hot Water	NaOH Solution
RS 300	4.51 ± 0.10 b	4.53 ± 0.11 b	6.77 ± 0.20 a	5.34 ± 0.07 a	3.91 ± 0.09 b	2.81 ± 0.02 c
RS 500	3.50 ± 0.25 b	4.16 ± 0.02 a	1.22 ± 0.02 c	9.50 ± 0.21 b	14.91 ± 0.79 a	8.39 ± 0.86 b
RS 700	0.21 ± 0.02 b	1.67 ± 0.14 a	0.30 ± 0.02 b	5.44 ± 1.10 c	34.00 ± 0.33 a	9.56 ± 0.78 b
ATB 300	1.93 ± 0.26 b	2.07 ± 0.01 b	2.94 ± 0.01 a	5.38 ± 0.11 a	4.42 ± 0.17 b	3.38 ± 0.05 c
ATB 500	3.16 ± 0.19 ab	2.26 ± 0.61 b	3.42 ± 0.45 a	4.50 ± 1.08 ab	5.26 ± 0.32 a	2.89 ± 0.03 b
ATB 700	1.81 ± 0.30 b	2.01 ± 0.71 ab	3.24 ± 0.47 a	3.83 ± 0.03 b	8.97 ± 0.74 a	2.13 ± 0.54 c
PS 300	1.59 ± 0.02 c	2.64 ± 0.13 b	3.52 ± 0.16 a	8.47 ± 0.44 a	4.49 ± 0.80 b	3.65 ± 0.19 b
PS 500	3.77 ± 0.55 a	3.26 ± 0.08 b	4.05 ± 0.51 a	5.58 ± 0.96 b	8.14 ± 0.22 a	3.36 ± 0.57 c
PS 700	1.64 ± 0.45 b	0.43 ± 0.13 c	3.42 ± 0.64 a	6.50 ± 0.01 b	8.71 ± 0.71 a	1.90 ± 0.10 c
Average	2.46 ± 0.24 b	2.56 ± 0.22 b	3.21 ± 0.27 a	6.06 ± 0.44 b	10.31 ± 0.46 a	4.23 ± 0.35 c

Different letters in the same line indicate significant differences (*p* < 0.05) among water, hot water and NaOH solution.

## Data Availability

Not applicable.

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
