# Peer review of "Investigating Biochar-Derived Dissolved Organic Carbon (DOC) Components Extracted Using a Sequential Extraction Protocol"

_materials, 2022, doi:10.3390/ma15113865_

Round 1

Reviewer 1 Report

The work is well organized and well described. The discussion and discussion of the results were well described and relevant works were cited. However, I have some comments and questions for the authors:

1) Too long abstract. Please shorten it and emphasize the novelty of this work.

2) Keywords are repeated from the title. Please change them to a different one.

3) Shorten your introduction a little. Some of the sentences in the DOC Paragraph and the next paragraph could fit better in the results discussion section.

4) Highlight the novelty. What is new in this research? How does it differ from the rest of the works? Based on the data presented in the introduction, this experience does not bring anything new.

5) Why was such biomass selected?

6) Add methodology of biomass analysis. Has any biochar analysis been performed as well? Ash content? Fixed carbon? FTIR? 

7) Add the company name to the TOC instrument and the UV-vis spectrophotometer

8) UV-Vis indicators. Add a citation of the work from which these indicators were taken.

9) A paragraph describing the parameters of biochar and the analysis of the impact of biomass and pyrolysis temperature of the tested biochars should be added. I think it is important and helpful for further discussion of the results.

10) Has the pH of the biomass or biochar been checked? Can this parameter affect the DOC?

Author Response

Response to the Review report 1

MS No. materials-1686742

Title: Investigating biochar-derived dissolved organic carbon (DOC) components extracted using a sequential extraction protocol

The work is well organized and well described. The discussion and discussion of the results were well described and relevant works were cited. However, I have some comments and questions for the authors:

Response: Thanks to the reviewer.

1) Too long abstract. Please shorten it and emphasize the novelty of this work.

Response: Thanks to the reviewer. Abstracts have been shortened. The novelty of this paper is that the use of a sequential extract protocol of water, hot water, and NaOH solution in combination with spectroscopic methods can successfully reveal the diversity of chemical structures and fractions of DOC that can be released from biochar. The DOC components of biochar could not be fully extracted with the same single solution of water, hot water and alkali. As described in Introduction “this study showed that using a sequential extract protocol of water, hot water, and NaOH solution in combination with spectroscopic methods can successfully reveal the diversity of biochar-derived components, which could lead to new insights for accurate assessment of potential environmental impact and new directions for biochar application.” in the revised manuscript.

2) Keywords are repeated from the title. Please change them to a different one.

Response: Correction has been done. Keywords has been changed, as “Keywords: biochar, DOC concentration. UV-vis analyses, PARAFAC analysis” in the revised manuscript.

3) Shorten your introduction a little. Some of the sentences in the DOC Paragraph and the next paragraph could fit better in the results discussion section.

Response: Thanks to the reviewer. Some descriptions have been deleted. Correction has been done.

4) Highlight the novelty. What is new in this research? How does it differ from the rest of the works? Based on the data presented in the introduction, this experience does not bring anything new.

Response: As mentioned above, the novelty of this paper is the use of a sequential extract protocol of water, hot water, and NaOH solution in combination with spectroscopic methods can successfully reveal the diversity of chemical structures and fractions of DOC that can be released from biochar. Few of researches have documented completely extracting the DOC generated from biochar using a stepwise process. It is critical to create a sequential extraction strategy based on the likely release sequence of DOC components to get a thorough extraction of DOC.  These findings are critical for predicting potential effects of biochar and its containing DOC on agriculture and soil environments.

5) Why was such biomass selected?

Response: Thanks to reviewer. Correction has been done as “Rape straw, apple tree branches and pine sawdust are selected as the feedstocks because they are widely distributed agricultural and forestry waste in China [25-27]”.

6) Add methodology of biomass analysis. Has any biochar analysis been performed as well? Ash content? Fixed carbon? FTIR?

Response: Thanks to the reviewer. The descriptions have been added, as “Proximate analysis of feedstocks were done by determine the content of moisture, ash, volatile matter, and calculating the content of fixed carbon by weight difference. Moisture content was determined by placing the samples into glass dishes being dried at 105 °C to a constant weight. The volatile matter was determined by keeping the samples into porcelain crucible in a muffle furnace at 900 ± 10 ℃for 7 min. Ash content was obtained by keeping samples into porcelain crucible in a muffle furnace at 700 ℃for half of an hour. The lignocellulosic content of samples, which includes hemicellulose, cellulose, and lignin was obtained based on the methods provided by Li et al. [28]. All the experimental determinations were repeated three times.” in the revised manuscript.

7) Add the company name to the TOC instrument and the UV-vis spectrophotometer

Response: We are very sorry for our negligence of the company name of TOC instrument and the UV-vis spectrophotometer. Correction has been made.

8) UV-Vis indicators. Add a citation of the work from which these indicators were taken.

Response: The citation work has been added.

9) A paragraph describing the parameters of biochar and the analysis of the impact of biomass and pyrolysis temperature of the tested biochars should be added. I think it is important and helpful for further discussion of the results.

Response: We appreciate the reviewer for this kind recommendation. It is really true as Reviewer suggested that biomass and pyrolysis temperature play an important role in the physicochemical properties of biochar. Wei et al. suggested that biochar produced at low temperature exhibited high yields, high dissolved organic carbon (DOC) content and unstable organic carbon content. In contrast, biochar formed at high temperature showed high C content and C stability with a low O/C and H/C ratios. In addition, the biochar pyrolyzed from pig manure contained the lowest DOC of the four biochar types (rice straw (RS), pine wood (PW), pig manure (PM) and sewage sludge (SS )[16]. However, our study focused directly on the effects of pyrolysis temperature and raw materials on biochar-derived DOC. For example, “RS is herbaceous biomass with a higher proportion of hemicellulose and cellulose ( as shown in table 1), which promotes the synthesis of volatile organic compounds and bio-oil inside the biochar pore structure by recondensation and trapping, which may then be released as DOC [37]. ATB and PS are woody materials that include a higher concentration of lignin components that are more difficult to breakdown, resulting in pyrolysis products with a higher solid content and a lower DOC content [36]. Herbaceous plants (e.g. RS) have greater ash content, catalysing the thermochemical reaction of lignocelluloses, resulting in increased DOC content [37]”, and “Increased pyrolysis temperature accelerated the carbonization process, resulting in the continuous decomposition, condensation, cyclisation, and polymerization of the labile organic compounds formed at lower pyrolysis temperatures and a decrease in DOC concentrations.” We are sorry. No correction was done.

10) Has the pH of the biomass or biochar been checked? Can this parameter affect the DOC?

Response: The pH value of biochar was measured by adding 2.5 g sample to 50 ml of deionized water as 1:20 (w/v), and the measured value varied between 6.83 to 9.84. while in the first step of sequential extraction experiments in this studies, 3 g of biochar was combined with 300 mL of deionized water as 1:100 (w/v). This corresponds to a 4-fold reduction in the amount of biochar applied compared to the amount measured for pH value. It can be assumed that the effect of pH of biochar is insignificant.

Author Response

Response to the Review report 2

MS No. materials-1686742

Title: Investigating biochar-derived dissolved organic carbon (DOC) components extracted using a sequential extraction protocol

The dissolved organic carbon is the most reactive fraction of organic matter. The sequential extraction of DOC and a characterization of its components it is very important to understand the behavior and contributions from biochar soil application. However, I think that the author should highlight the novelty of the paper, which is not so clear. Compare different biochars DOC fractions and composition is interest and important to understand which one can contribute better to improve soil fertility, for example. However, the way that the discussion was conducted could be improved showing not only the differences on composition but also suggesting which one is better for soil amendment. These things should be pointed to show the relevance of the study. Further, the discussion with PARAFAC is confusing, and the way that the results were presented make even difficult to follow and to establish a comparison among the biochars. I strongly recommend the review all over the paper. The discussion can be improved making a comparison with other protocols available on literature regarding DOC extraction and composition. I also recommend a review of the English language.

Response: Thanks to the reviewer. The novelty of this paper is the use of a sequential extract protocol of water, hot water, and NaOH solution in combination with spectroscopic methods can successfully reveal the diversity of chemical structures and fractions of DOC that can be released from biochar. DOC produced from biochar extracted conventionally using acid (0.1 M HCl), base (0.1 M NaOH), salt solutions (0.1 mol/L), NaCl deionized water, and hot water (60 or 80 °C). The difference of DOC characteristics in different extraction solutions indicates the diversity of DOC components. It is also indicated that a specific extraction solution can only extract certain DOC compounds from biochar. Thus, drawing quantitative and qualitative conclusions about the diversity of compounds and molecular characteristics of all potentially releasable biochar-derived DOC fractions based on water-, hot water-, salt-, acid- or alkali-extractable components alone is problematic. To deal with this issue, many researchers then use sequential extraction procedure or multi-step extraction methods, and the results shows these methods allows for revealing the diversity of organic matter components [22-25]. However, few of researches have documented completely extracting the DOC generated from biochar using a stepwise process. It is critical to create a sequential extraction strategy based on the likely release sequence of DOC components to get a thorough extraction of DOC. These findings are critical for predicting potential effects on agriculture and soil environments. As Reviewer suggested that “compare different biochars DOC fractions and composition is interest and important to understand which one can contribute better to improve soil fertility”, which is very helpful improving our paper and guiding significance our future researches. However, in this paper, we focused on comparing the differences in concentration, chemical structure and component fractions of DOC extracted from water, hot water and NaOH, aiming to illustrate the diversity of biochar-derived DOC fractions.

1) line 10, page 1: “Biochar-derived dissolved organic carbon (DOC) as the most important component of biochar can be released in farmland fertility improvement, soil amendment, and soil remediation, and the complexity of molecular structures and diversity of DOC compounds have influenced these functions to some extent”. The sentence is confusing, please, rewrite the phrase adding comas and check the verb tense. Please, check if the follow sentence agrees with your purpose. “Biochar-derived dissolved organic carbon (DOC), as the most important component of biochar, can be released in farmland improving fertility, acting on soil amendment and remediation. The complexity of molecular structures and diversity of DOC compounds have influenced these functions to some extent.”

Response: Thanks to the reviewer. Corrections have been made.

2) Abstract: as the authors made a sequential extraction using water, hot-water and NaOH (as they mention on line 13), I think they should follow the same sequence to explain the DOC extracts composition. Doing this, it is easier to readers understanding once follow the same sequence of methodology to explain the results.

Response: Correction has been done.

3) Line 54, page 2: “change in DOC composition to be more aromatic” – Please, check the sentence to improve the meaning. There is a release of aromatic compounds what will contribute to change the content of aromatic compounds in DOC, right?

Response: We have made correction according to the Reviewer’s comments. “For example, combining cacao shell-derived biochar with acrisols at 10% wt. resulted in a 15-fold increase in soil DOC content and changed the DOC composition towards molecules with a larger size and higher aromaticity

4) Line 75, page 2: “which are more conductive to DOC formation” – I think “conductive” it is not the best word, maybe “lead” is better. Please, think about.

Response: Thanks to the reviewer. We have made correction according to the Reviewer’s comments. “conductive” has been changed to “lead”.

5) Line 80, page 2: “climbed” is not the best word, you can say increase.

Response: Thanks to the reviewer. Correction has been done. “climbed” has been changed to “increase”

6) Line 82, page 2: “humic-like compounds increased and gradually decreased” – Increase until which temperature? And from which temperature the humic-like substances star to decrease? I think this information is important.

Response: Thanks to the reviewer. We have made correction according to the Reviewer’s comments. “but the humic acid-like compounds increased until 400 °C and then gradually decreased [20].”

7) Line 108-109, page 2: “indicates the diversity of DOC components, it is strongly dependent on the physicochemical” – please change to “indicates the diversity of DOC components, which it is strongly dependent on the physicochemical”.

Response: Thanks to the reviewer. We have re-written this part according to the Reviewer’s suggestion.

8) Line 179, page 4: What the authors means by “underwent three parallel trials”? Did they do three replicates for each extraction?

Response: Thanks to the reviewer. We are sorry for the ambiguous description, “underwent three parallel trials” refers as “do three replicates for each extraction”. Correction has been done.

9) Line 197, page 5: What about C in the equation 4? Please check and insert the meaning (DOC concentration) on the description.

Response: Thanks to the reviewer. “C is the TOC concentration (mg·L-1) in the extraction solution” as shows Line 202, page 5, so it is omitted in the discussion below.

10) Line 215, page 5: Once the values are in the table 2, it is not necessary to show all of them as it is in the text, because it is a little bit difficult to understand and stablish any comparison.

Response: Thanks to the reviewer. As Reviewer suggested that the values listed in the text have been deleted.

11) Line 229, page 5: “the pH of the soil may have a significant role in the amendment's efficacy”. – There is an extensive discussion in literature about the relationship of biochar pH and soil pH. I think that this discussion should be improved considering which soil pH could contribute for the releasing of DOC from biochar, or which ones might not.

Response: Thanks to the reviewer. According to the results of our literature survey, the pH of saline-alkali soil can significantly promote the release of DOC from corn straw biochar [33], while the pH of Gray Brown Luvisol had no significant effects on the release of DOC from spent coffee grounds, wood pellets, and horse bedding compost biochars [Zhang et al. Effects of biochar amendments on soil microbial biomass and activity. Journal of Environmental Quality, 2014, 43, 2104-2114]. A minor correction was been done as “the pH of the soil ( especially for soda saline-alkali soil ) may have a significant role in the amendment's efficacy [31-33]

12) lines 231-239, page 6: The authors mention some compounds and affirm that they are presents in their DOC extracts (water and hot water). Were these compounds identified in this work? It seems that is made a correlation with other works, so how can the authors affirm that they are presents in their extracts?

Response: Thanks to the reviewer. These compounds are not identified in this work but cited from reference [22]. The reason why their presence in aqueous solution can be confirmed is that the extractants are the same and the samples being extracted have some similarity, Therefore, we consider these compounds are presents in their extracts.

13) Table 2, page 6: “Different letters with a column indicate significant differences (p<0.05) among water, hot water and 263 NaOH solution.” – Different letters in the same line indicates significant differences, right? Not in a column. Please check the sentence, once the statistical differences were analyzed among water, hot water and NaOH for the same sample.

Response: Thanks to the reviewer. We are very sorry for our incorrect writing. Correction has been done. “Different letters in the same line indicates significant differences (p<0.05) among water, hot water and NaOH solution.

14) Line 268, page 7: “The relative quantities of water and hot water extractable fractions declined as the pyrolysis temperature rose, but the NaOH solution extractable fractions increased (Figure 1). – Please, check the sentence. This is not in accordance with all samples, just for RS biochar.

Response: Thanks to the reviewer. We are sorry for the nonsence description. Correction has been done.

15) Line 274, page 7: “Please, check the sentence. This is not in accordance with all samples, just for RS biochar.” – This sentence doesn’t make sense. Please remove it, or rewrite. Once UV-Vis and fluorescence were made, you can just mention that the investigation regarding DOC composition was made and indicate where these results are showed.

Response: Thanks to the reviewer. We are sorry for the ambiguous description. As Reviewer suggested that “The differences in the chemical structure characteristics of DOC components need to be investigated further using UV–vis and fluorescence, as described below.” has been deleted.

16) Line 283, page 7: “with rising the pyrolysis” correct to “rising the pyrolysis”.

Response: Thanks to the reviewer. We have made correct according the Reviewer’s comments.

17) Line 282, page 7: The value of SUVA254 is showed as L mg-1·m-1 , but it is, generally, presented in function of carbon concentration, isn’t it? Please check.

Response: Thanks to the reviewer. As illustrated in literature 38, “SUVA254 is defined as the UV absorbance at 254 nanometers measured in inverse meters (m-1) divided by the DOC concentration measured in milligrams per liter (mg·L-1).” The value of SUVA254 presented in function of carbon concentration and UV absorbance at 254 nanometers.

18) Line 296, page 7: “Additionally, the hydrophobic components of biochar can result in increased percentages of entrapped air, reducing the amount of available soil water after delivered to the soil”. Please, could the authors explain better this sentence? Available soil water? Are the authors making reference to the formation of aggregates due interaction of hydrophobic portion of biochar with soil?

Response: Thanks to the reviewer. The hydrophobic fraction of organic matter has an important influence on the soil aggregates. Soil aggregate stability was improved and maintained with time more by hydrophobic than by hydrophilic components of organic matter. Long-lasting aggregate stability of soils can be thus achieved by addition of hydrophobic humic material with hydrophobic organic wastes. The hydrophobic components of biochar can result in increased increase the long-term agglomeration stability of the soil. The description and explanation have been added “Once the hydrophobic organic matter component of biochar enters the soil, it promotes the long-term stability of soil aggregates due to the formation of organic matter-soil complexes with hydrophobic characteristics [44]” in the revised manuscript.

19) Line 297, page 7: The authors mention the increase in hydrophobicity due high temperature pyrolysis of RS. However, at line 282-286 they said that values of SUVA254 < 3 represents more aromatic hydrophilic substances in DOC, and once the SUVA254 decreased with increasing temperature for RS in all extracts, the RS hydrophilicity is increasing, right? Please, check.

Response: Thanks to the reviewer. We are sorry for the misdescription. We have made correction according the Reviewer’s comments. “The DOC portion of RS biochar generated at a high temperature (700 °C) demonstrates increased hydrophilicity, which affects the amount of metals and organic pollutants co-sorption and co-transport in soil solution [45]

20) Line 322, page 8:“The E2/E3 ratio of RS biochar-derived DOC in water, hot water, and NaOH increased from 5.34 ± 0.07 to 9.50 ± 0.21, 3.91 ± 0.09 to 34.00 ± 0.33, and 2.81 ± 0.02 to 9.56 ± 0.78 with the rising of pyrolysis temperature from 300 to 700 °C respectively.”-Please, check the use of “respectively”. The values for RS at 500 and 700 °C, because in the table the value 9.5 correspond to 500 °C and not 700 °C.

Response: Thanks to the reviewer. We are sorry for the ambiguous description. Correction has been made.

21) line 342-347, page 8: The values discussed in this paragraph are from the line “average” in table 3. What is the meaning to make an average among the values of different samples (biochar)? Is it real representative to discuss? I don’t think that it is right, once you are comparing vary different values (from 5 to 34) and samples (three biochars), that shows different information if analyzed separately.

Response: Thanks to the reviewer. The differences in the average value among the different columns represent the variability between groups. Multiple comparison analysis in 2-way ANOVA can be used to compare the main column effect. Namely, it can be used to compare the E2/E3 variability of water, hot water and NaOH extracted fraction DOC, and then to reveale the molecular size of biochar-derived DOC released in water, hot water and NaOH solution . We are sorry. No correction was done.

22) Line 355, page 9: “Additionally, the C3 component exhibited two peaks with a maximum excitation at 230 and 356 maximum emission at 350 and 460 nm.”. – The C3 component at Figure 2 shows just one component, please check the Figure and the text information. Further, as mentioned at line 357 “C3 has a longer emission wavelength than C2, indicating that it contains more aromatic”, this happens to C2 component, not C3. Please, check this information too.

Response: Thanks to the reviewer. We are sorry for the ambiguous description. Correction has been made.

23) Figure 2: The PARAFAC components presented on figure 2 were obtained from which sample and which extractant? Even they are just a representative of the majority components found in the DOC fractions, the sample that they were obtained should be described at least in the figure description.

Response: Thanks to the reviewer. The PARAFAC components presented on figure 2 were obtained from RS, ATB and PS biochars released in the water, hot water and NaOH solution. The key to obtaining PARAFAC components is to perform Split Half analysis and validation for fluorescence intensity of all samples. The PARAFAC of the EEM for all samples in this paper was based entirely on the method provided in the literature [29], as described “The obtained EEM data were analysed using PARAFAC modelling in MATLAB R2010a, as described by Stedmon and Bro [29]

24) Discussion on page 10: The authors should explain how they calculated the distribution relative expressed in percentage regarding the contributions of PARAFC components in the extracts. I can not understand how they estimate the percentages. Whit PARAFAC you can estimate the components, but also the amount of each one in the sample?

Response: Thanks to the reviewer. Supplement has been done as “Based on PARAFC process for analyzing the EEM fluorescence of the samples, once the PARAFAC model has identified the components, the positions of the fluorescence peaks at different excitation and emission wavelengths and the corresponding fluorescence intensities of the components contained in the sample are automatically output. Relative proportions of different components can be calculated based on their fluorescence intensity.

25) Line 384, page10: “the presence of OH- (NaOH) may decrease the hydrogen connection between the humic-like substance and the biochar, encouraging humic-like substance release from the biochar.” The meaning of this sentence is not clear, please rewrite and I encourage the authors to take care with the words that they use sometimes in the text as climb (other parts in the text) and “encourage” in the sentence mentioned.

Response: Thanks to the reviewer. Correction has been made. The description have been added “the presence of OH- (NaOH) may weaken the hydrogen connection between the humic-like substance and the biochar, prompting the release of humic substances from the biochar.” in the revised manuscript.

26) Page 10: It is very hard to follow the discussion just using the Figure 3. It is impossible to check the information if the authors don’t provide the figures (even as support information) of EEM for all samples. I think that these materials should be presented for a better evaluation of the results. All discussion is made based on the percentages presented on Figure 3, however the explanation about how they were calculated were not provided.

Response: Thanks to the reviewer. Due to the large number of samples in this study (27 samples), each sample corresponds to a 3D fluorescence map. Inserting all maps into the document will bring inconvenience in reading and layout. As responded to Q24, supplement has been done as “Based on PARAFC process for analyzing the EEM fluorescence of the samples, once the PARAFAC model has identified the components, the positions of the fluorescence peaks at different excitation and emission wavelengths and the corresponding fluorescence intensities of the components contained in the sample are automatically output. Relative proportions of different components can be calculated based on their fluorescence intensity.

27) The information obtained from UV-Vis could be correlated with the results from PARAFAC to support the observations provided on UV-Vis result section. I encourage the authors to think about to connect the results, once they might support each other specially when they make some comments regarding DOC composition on UV-Vis just comparing with other works.

Response: We appreciate the reviewer for this kind recommendation. As far as we know, UV–Vis absorption spectroscopy is the measurement of the attenuation of a beam of light after it passes through a sample or after reflection from a sample surface. The different wavelengths are believed to identify different chromophores of DOC. Absorbance at specific wavelengths (254 nm), the ratio between the absorbance of different wavelengths (E2/E3), or the UV absorbance of a given sample at 254 nm divided by the DOC concentration (SUVA254), which can provide aromaticity, molecular weight, hydrophilic and hydrophobic characteristics of the sample. The EEM spectra visualises a range of different fluorophores covering the excitation and emission wavelengths range from 200 nm to 500 nm. Multivariate analyzes of EEM spectra using PARAFAC analysis have become the favored approach for investigating detailed fluorescence differences among samples. Combining the UV-Vis results with the PARAFAC analysis results can provide a new perspective on the chemical structure characterization of DOC. However, the small number and variety of samples in this study may lead to less accurate analytical results. In future studies, we can combine the UV-Vis results with the PARAFAC analysis results to form comprehensive understanding of the chemical structure and component characteristics of DOC components.

Reviewer 3 Report

Comments

The dissolved organic carbon is the most reactive fraction of organic matter. The sequential extraction of DOC and a characterization of its components it is very important to understand the behavior and contributions from biochar soil application. However, I think that the author should highlight the novelty of the paper, which is not so clear. Compare different biochars DOC fractions and composition is interest and important to understand which one can contribute better to improve soil fertility, for example. However, the way that the discussion was conducted could be improved showing not only the differences on composition but also suggesting which one is better for soil amendment. These things should be pointed to show the relevance of the study. Further, the discussion with PARAFAC is confusing, and the way that the results were presented make even difficult to follow and to establish a comparison among the biochars. I strongly recommend the review all over the paper. The discussion can be improved making a comparison with other protocols available on literature regarding DOC extraction and composition. I also recommend a review of the English language.

1) line 10, page 1: “Biochar-derived dissolved organic carbon (DOC) as the most important component of biochar can be released in farmland fertility improvement, soil amendment, and soil remediation, and the complexity of molecular structures and diversity of DOC compounds have influenced these functions to some extent”. - The sentence is confusing, please, rewrite the phrase adding comas and check the verb tense. Please, check if the follow sentence agrees with your purpose. “Biochar-derived dissolved organic carbon (DOC), as the most important component of biochar, can be released in farmland improving fertility, acting on soil amendment and remediation. The complexity of molecular structures and diversity of DOC compounds have influenced these functions to some extent.”

2) Abstract: as the authors made a sequential extraction using water, hot-water and NaOH (as they mention on line 13), I think they should follow the same sequence to explain the DOC extracts composition. Doing this, it is easier to readers understanding once follow the same sequence of methodology to explain the results.

3) Line 54, page 2: “change in DOC composition to be more aromatic” – Please, check the sentence to improve the meaning. There is a release of aromatic compounds what will contribute to change the content of aromatic compounds in DOC, right?

4) Line 75, page 2: “which are more conductive to DOC formation” – I think “conductive” it is not the best word, maybe “lead” is better. Please, think about.

5) Line 80, page 2: “climbed” is not the best word, you can say increase.

6) Line 82, page 2: “humic-like compounds increased and gradually decreased” – Increase until which temperature? And from which temperature the humic-like substances star to decrease? I think this information is important.

7) Line 108-109, page 2: “indicates the diversity of DOC components, it is strongly dependent on the physicochemical” – please change to “indicates the diversity of DOC components, which it is strongly dependent on the physicochemical”.

8) Line 179, page 4: What the authors means by “underwent three parallel trials”? Did they do three replicates for each extraction?

9) Line 197, page 5: What about C in the equation 4? Please check and insert the meaning (DOC concentration) on the description.

10) Line 215, page 5: Once the values are in the table 2, it is not necessary to show all of them as it is in the text, because it is a little bit difficult to understand and stablish any comparison.

 11) Line 229, page 5: “the pH of the soil may have a significant role in the amendment's efficacy”. – There is an extensive discussion in literature about the relationship of biochar pH and soil pH. I think that this discussion should be improved considering which soil pH could contribute for the releasing of DOC from biochar, or which ones might not.

12) lines 231-239, page 6: The authors mention some compounds and affirm that they are presents in their DOC extracts (water and hot water). Were these compounds identified in this work? It seems that is made a correlation with other works, so how can the authors affirm that they are presents in their extracts?

13) Table 2, page 6: “Different letters with a column indicate significant differences (p<0.05) among water, hot water and 263 NaOH solution.” – Different letters in the same line indicates significant differences, right? Not in a column. Please check the sentence, once the statistical differences were analyzed among water, hot water and NaOH for the same sample.

14) Line 268, page 7: “The relative quantities of water and hot water extractable fractions declined as the pyrolysis temperature rose, but the NaOH solution extractable fractions increased (Figure 1). – Please, check the sentence. This is not in accordance with all samples, just for RS biochar.

15) Line 274, page 7: “Please, check the sentence. This is not in accordance with all samples, just for RS biochar.” – This sentence doesn’t make sense. Please remove it, or rewrite. Once UV-Vis and fluorescence were made, you can just mention that the investigation regarding DOC composition was made and indicate where these results are showed.

16) Line 283, page 7: “with rising the pyrolysis” correct to “rising the pyrolysis”.

17) Line 282, page 7: The value of SUVA254 is showed as L mg-1 m-1 , but it is, generally, presented in function of carbon concentration, isn’t it? Please check.

18) Line 296, page 7: “Additionally, the hydrophobic components of biochar can result in increased percentages of entrapped air, reducing the amount of available soil water after delivered to the soil”. Please, could the authors explain better this sentence? Available soil water? Are the authors making reference to the formation of aggregates due interaction of hydrophobic portion of biochar with soil?

19) Line 297, page 7: The authors mention the increase in hydrophobicity due high temperature pyrolysis of RS. However, at line 282-286 they said that values of SUVA254 < 3 represents more aromatic hydrophilic substances in DOC, and once the SUVA254 decreased with increasing temperature for RS in all extracts, the RS hydrophilicity is increasing, right? Please, check.

20) Line 322, page 8: “The E2/E3 ratio of RS biochar-derived DOC in water, hot water, and NaOH increased from 5.34 ± 0.07 to 9.50 ± 0.21, 3.91 ± 0.09 to 34.00 ± 0.33, and 2.81 ± 0.02 to 9.56 ± 0.78 with the rising of pyrolysis temperature from 300 to 700 °C respectively.”- Please, check the use of “respectively”. The values for RS at 500 and 700 °C, because in the table the value 9.5 correspond to 500 °C and not 700 °C.

21) line 342-347, page 8: The values discussed in this paragraph are from the line “average” in table 3. What is the meaning to make an average among the values of different samples (biochar)? Is it real representative to discuss? I don’t think that it is right, once you are comparing vary different values (from 5 to 34) and samples (three biochars), that shows different information if analyzed separately.

22) Line 355, page 9: “Additionally, the C3 component exhibited two peaks with a maximum excitation at 230 and 356 maximum emission at 350 and 460 nm.”. – The C3 component at Figure 2 shows just one component, please check the Figure and the text information. Further, as mentioned at line 357 “C3 has a longer emission wavelength than C2, indicating that it contains more aromatic”, this happens to C2 component, not C3. Please, check this information too.

23) Figure 2: The PARAFAC components presented on figure 2 were obtained from which sample and which extractant? Even they are just a representative of the majority components found in the DOC fractions, the sample that they were obtained should be described at least in the figure description.

24) Discussion on page 10: The authors should explain how they calculated the distribution relative expressed in percentage regarding the contributions of PARAFC components in the extracts. I can not understand how they estimate the percentages. Whit PARAFAC you can estimate the components, but also the amount of each one in the sample?

25) Line 384, page10: “the presence of OH- (NaOH) may decrease the hydrogen connection between the humic-like substance and the biochar, encouraging humic-like substance release from the biochar.” – The meaning of this sentence is not clear, please rewrite and I encourage the authors to take care with the words that they use sometimes in the text as climb (other parts in the text) and “encourage” in the sentence mentioned.

26) Page 10: It is very hard to follow the discussion just using the Figure 3. It is impossible to check the information if the authors don’t provide the figures (even as support information) of EEM for all samples. I think that these materials should be presented for a better evaluation of the results. All discussion is made based on the percentages presented on Figure 3, however the explanation about how they were calculated were not provided.

27) The information obtained from UV-Vis could be correlated with the results from PARAFAC to support the observations provided on UV-Vis result section. I encourage the authors to think about to connect the results, once they might support each other specially when they make some comments regarding DOC composition on UV-Vis just comparing with other works.

Author Response

Response to the Review report 3

MS No. materials-1686742

Title: Investigating biochar-derived dissolved organic carbon (DOC) components extracted using a sequential extraction protocol

The article is devoted to the study of the composition and amount of organic matter biochar obtained at different temperatures from different substrates. At the same time, the extraction of organic matter occurs in three different ways: in water, hot water, and alkali. The design of the experiment is not in doubt, but it remains unclear what connection the extraction of biochar organic matter with hot water and alkali has with its decomposition in natural conditions: water at an ambient temperature of no more than 30-40 degrees Celsius and the absence of strong alkali. At the same time, it was not considered which organic fractions will get from the biochar into the soil at a lower temperature or with a slightly acidic or slightly alkaline reaction of the medium, which are much more common in the soil? how biochar organic matter will decompose under the influence of bacteria and oxygen (main components of organic matter decomposition of soil)? All these parameters require at least not research but discussion. Because it is under such conditions that biochar decomposes in the soil, and not in those that were modeled in the laboratory as part of the current work.

Response: Thanks to the reviewer. It is really true as Reviewer suggested that water at an ambient temperature of no more than 30-40 degrees Celsius and the absence of strong alkali. However, as previous studies reported, the release quantity and quality of DOM from biochar were significantly influenced by the environmental pH and temperature. Water-extractable DOC increased in an exponential manner between 10 and 80°C. There was relatively little change in dissolved C between 10 and 30 °C, but a large increase above 30°C. This may be due to higher extraction temperature (50, 60 or 80 °C) [19, 20, 68] favorably lead to higher DOC solubility and the desorption of higher fixed carbon content from the biochar. These fractions are mostly carbohydrates, phenols, lignin monomers, and nitrogen-containing chemicals [32, 35] and exhibit a stronger biodegradability than cold water extracts. Therefore, setting the temperature of hot water extraction to 80 °C is helpful to comprehensively collect the hot water extraction components from biochar and evaluate the bioavailability of biochar-derived DOC.

Alkaline extraction is widely used as a method for extracting organic matter contained in particles such as soil, sediment. The most common extraction procedures involve the use of alkaline solutions particularly NaOH, which is more efficient than KOH. The concentration of the alkali used is important in determining the amount of humic substance extracted. For example, 0.1 M NaOH is a more efficient extractant than higher concentrations (0.5 M) [Vaughan, D. et al. Introduction soil organic Matter—a perspective on its nature, extraction, turnover and role in soil fertility. In Soil organic matter and biological activity (pp. 1-35). 1985, Springer, Dordrecht.]. At this pH (0.1 M), most oxygen-containing functional groups in organic matter are ionized, making organic compounds bearing such groups much more soluble in water. Therefore, some studies have attempted to extract the fractions of dissolved organic carbon from biochar using low concentrations of alkali solutions. Bakshi et al., found that 0.05 M NaOH treatment was the most effective in extracting liable carbon from biochar, compared with other concentrations [Bakshi et al. Quantification and characterization of chemically-and thermally-labile and recalcitrant biochar fractions. Chemosphere, 2018, 194, 247-255]. Uchimiya et al., investigated the chemical structure of biochar’s labile dissolved organic carbon (DOC) by using hot water (80 â—¦C) and then cold (room temperature) 0.05 M NaOH sequential extraction methods. Wei et al., also investigated the effect of the pyrolysis temperature and feedstock on the aromaticity of biochar-derived DOC by using cold water (25 â—¦C), hot water(80 â—¦C), and alkali sequential (0.05 M NaOH) extraction methods [68].

As “it was not considered which organic fractions will get from the biochar into the soil at a lower temperature or with a slightly acidic or slightly alkaline reaction of the medium, which are much more common in the soil? how biochar organic matter will decompose under the influence of bacteria and oxygen (main components of organic matter decomposition of soil)?” mentioned by the Reviewer, which is very helpful improving our paper and guiding significance our future researches. The release of DOC from biochar in soil is a complex process influenced by many factors, such as physicochemical properties of biochar and soil, type and texture of the soil, climatic conditions and duration of application. The decomposition of biochar-derived DOC is also influenced by the factors of pH, water content and temperature of soil, bacteria fungi and oxygen, etc. However, in this paper, we focused on comparing the differences in concentration, chemical structure and component fractions of DOC extracted from water, hot water and NaOH, aiming to illustrate the diversity of biochar-derived DOC fractions. The aim is to provide a theoretical basis for the potential impact of biochar in predicting agricultural and soil environments.

1) Investigating biochar-derived dissolved organic carbon (DOC) components extracted using a sequential extraction protocol. ( Reviewer's note: The title of the article and the keywords are practically the same. Perhaps one of these paragraphs should detail the extraction protocol (water, hot water, and lye) or the substrate (Rape straw, apple tree branches, and pine sawdust (PS))

Response: Correction has been done. Keywords has been changed, as “Keywords: biochar, DOC concentration. UV-vis analyses, PARAFAC analysis” in the revised manuscript.

2) The basic composition of samples ( Reviewer's note: Please add a column “n” to show the number of samples based on which the analysis was performed)

Response: Thanks to the reviewer. “n=3” has been added.

3) The spectral absorption ratio of 254 and 365 nm (E2/E3) was calculated using the following equation (2):?2/?3=?254/?365 (2) ( Reviewer's note: If ?254 is an absorption at a certain wavelength range, then the ratio of A254 and A365 should be a numerical value, denoted by a letter, and not the ratio of E2 to E3. In any case, it is not clear what E2/E3 are?)

Response: Thanks to the reviewer. Absorbance values at 250 and 365nm were recorded for the calculation of E2/E3, and for estimating the aromaticity of the solutes, respectively cited from the literature [47].

4) The specific UV absorption at 254 nm (SUVA254) ( Reviewer's note: Dimension units must be given for ?254 and SUVA254)

Response: Thanks to the reviewer. Correction has been done.

5)“Each experiment was replicated three times…”( Reviewer's note: What is meant by the word “experiment”? - extraction of carbon fractions or their analysis on a UV-vis analyzer or on a PARAFAC or both? Please indicate “n” for each step so that the experiment can be reproduced.)

Response: Thanks to the reviewer. “experiment” in “Each experiment was replicated three times…” means “extraction”. Correction has been done.

6) “These fractions comprised the widest variety of chemicals, from carbohydrate-like to amino-sugar-like to polyphenolic compounds.” ( Reviewer's note: If the factions are known exactly, list them, and if they are not known, it does not make sense to list them in such a general way.)

Response: Thanks to the reviewer. Correction has been done.

7) Table 2 ( Reviewer's note: Please add a column “n” to show the number of samples based on which the analysis was performed;)

Response: Thanks to the reviewer. “n=3” has been added.

8) Figure 1.( Reviewer's note: Check “ATP” please)

Response: Thanks to the reviewer. We are sorry for the misdescription. Correction has been made.

9) The quantities of water, hot water, and sodium hydroxide in ATB and PS biochar are diametrically opposed to those in RS biochar .( Reviewer's note: What does "diametrically opposed" mean? Has any statistical test been carried out? The term "diametrically opposite" should perhaps be replaced by another one)

Response: Thanks to the reviewer. Correction has been made. “diametrically opposed” has been changed to “completely opposed”

9) Table 3 ( Reviewer's note: Please add a column “n” to show the number of samples based on which the analysis was performed;)

Response: Thanks to the reviewer. “n=3” has been added.

10) Conclusions (Perhaps a conclusion in the conclusion section, based on the information shown in Tables 2 and 3, is that the higher the temperature of the biochar during biochar production, the more valuable organic compounds appear to be volatilized or condensed or otherwise lost during production. Perhaps, if the authors agree with this assumption, the most optimal biochar production temperature is 300 or 500 degrees, but not 700. At the same time, it seems that the RS 300 sample is the richest in organic matter of various fractions.)

Response: Thanks to the reviewer. It is really true as Reviewer suggested that RS 300 sample is the richest in organic matter of various fractions. However, RS 300 has a larger molecular weight, stronger aromaticity and hydrophobicity. And “The hydrophobic component of DOC was highly prefered over soil binding, resulting in decreased DOC-induced transport of metals and organic contaminants but increased sorption to the mineral phase [43].” Therefore, in practical applications, depending on the specific purpose, it is important to consider not only the potential DOC concentration released from the biochar, but also the DOC component characteristics when selecting the biochar. In addition, in this paper, we focused on comparing the differences in concentration, chemical structure and component fractions of DOC extracted from water, hot water and NaOH, aiming to illustrate the diversity of biochar-derived DOC fractions. We are sorry. No correction was made.

Round 2

Reviewer 1 Report

I recommend that the revised manuscript can be approved for publication.